# Interedge backscattering in time-reversal symmetric quantum spin Hall Josephson junctions

Cajetan Heinz,[1] Patrik Recher,[1,2] and Fernando Dominguez[1,3]

[1]*Institut für Mathematische Physik, Technische Universität Braunschweig, D-38106 Braunschweig, Germany*
[2]*Laboratory for Emerging Nanometrology Braunschweig, D-38106 Braunschweig, Germany*
[3]*Faculty of Physics and Astrophysics and Würzburg-Dresden Cluster of Excellence ct.qmat,*
*University of Würzburg, 97074 Würzburg, Germany*
(Dated: October 17, 2024)

Using standard tight-binding methods, we investigate a novel backscattering mechanism taking place on quantum spin Hall N'SNSN' Josephson junctions in the presence of time-reversal symmetry. This extended geometry allows for the interplay between two types of Andreev bound states (ABS): the usual phase-dependent ABS localized at the edges of the central SNS junction *and* phase-independent ABS localized at the edges of the N'S regions. Crucially, the latter arise at discrete energies $E_n$ and mediate a backscattering process between opposite edges on the SNS junction, yielding gap openings when both types of ABS are at resonance. In this scenario, a $4\pi$-periodic ABS decouples from the rest of the spectrum, and thus, it can be probed preventing the emission to the quasicontinuum. Interestingly, this backscattering mechanism introduces a new length scale, determining the ratio between $4\pi$- and $2\pi$-periodic supercurrent contributions and distorts the superconducting quantum interference (SQI) pattern. Finally, to proof the participation of these ABS, we propose to use a magnetic flux to tune $E_n$ to zero, resulting in the selective lifting of the fractional Josephson effect.

Josephson junctions (JJs) based on a single proximitized quantum spin-Hall (QSH) edge [1, 2] reveal their topological character by the presence of zero energy Majorana bound states for the superconducting phase difference $\phi = \pi$. Under parity conservation, the Josephson frequency is halved with respect to the conventional one, $f = 2eV/h \rightarrow eV/h$. The so-called *fractional Josephson effect* can, thus, be detected by probing observables linked to the periodicity of the supercurrent, e.g. the Shapiro experiment, developing constant voltage steps cleaved at $V_n = n\hbar\omega/2e$ ($V_n = n\hbar\omega/e$) for conventional (topological) JJs, with $\omega$ an external frequency, or measuring the Josephson radiation, with frequency $f$ or $f/2$ in conventional or topological JJs.

There is however a practical problem that one needs to circumvent to measure the fractional Josephson effect: in the presence of time-reversal symmetry (TRS), single-edge ABS are protected against backscattering, yielding a spectrum with no gap openings. Hence, a driven scenario leads irrevocably to an exchange of particles with the quasicontinuum, with the consequent change of parity and the destruction of the $4\pi$-periodicity.

Unfortunately, opening sizeable gaps by breaking TRS, i.e. adding an external magnetic field or magnetic add atoms, is technically difficult and can lead to unwanted phenomena, such as screening currents on the superconductor. Indeed, experiments performed so far in quantum spin-Hall Josephson junctions are under TRS conditions [3–8]. Unexpectedly, two of these experiments have shown signatures compatible with a topological ground state, with the absence of odd Shapiro steps [5] and the measurement of the fractional Josephson frequency $f/2$ in the Josephson radiation [6]. Previous theoretical

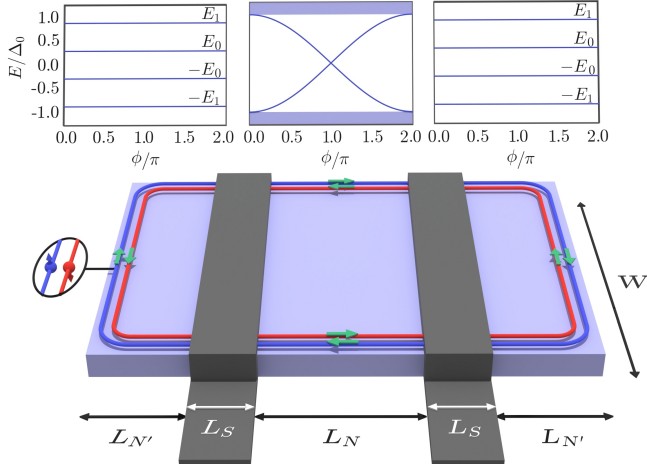

FIG. 1. Sketch of the extended quantum spin Hall Josephson junction. Here, the partial cover of the QSH bar (light blue) by superconducting leads (black) defines the extended N'SNSN' Josephson junction. Helical edge states are represented by blue and red curves. Inset: Andreev spectra as a function of the phase difference $\phi$ of the disconnected parts N'S (left), SNS (center) and SN' (right) regions.

works have made these experimental findings compatible with the absence of an explicit TRS breaking mechanism, by either considering a two-particle backscattering with large dissipation [9] or from a retardation effect present both in trivial and in topological superconductors [10]. Alternatively, one could attribute the $4\pi$-periodicity to a trivial scenario with non-adiabatic transitions between Andreev bound states [11–18] or by the presence of an environmental parasitic impedance [19].

In this contribution, we investigate a way to isolate energetically the topological ABS from the quasicontinuum without breaking time-reversal symmetry. To this aim, we engineer a backscattering process between opposite edges, mediated by an additional ABS present in the extended N'SNSN' junction, see Fig. 1 [20]. In this geometry, the central SNS junction is embedded in two additional normal N' parts resulting, for example, from the partial covering of the QSH bar with superconducting fingers, see Fig. 1. Here, two types of ABS arise: the usual phase-dependent ABS [1, 2] localized at the edges of the SNS region, and a phase-independent ABS localized at the N'S regions. The latter exhibits a discrete energy spectrum determined by the perimeter of the N' regions, namely

$$E_n = \frac{\hbar v_F}{p_{N'}} \pi (n + 1/2), \qquad (1)$$

with $n \in \mathbb{Z}$ and $p_{N'} = 2L_{N'} + W$, see Fig. 1. Interestingly, when the width of the superconducting fingers is smaller or of the order of the superconducting coherence length ($L_s \lesssim \xi_s$), both types of ABS hybridize when they are in resonance. In this scenario, avoided level crossings develop, yielding a $4\pi$-periodic ABS decoupled from the rest of the spectrum. We explore further consequences of the interplay of this new length scale $L_{N'}$ and its associated energy $E_0$ with the rest of the parameters of the junction, finding a modified ratio of the $4\pi$- and $2\pi$-periodic critical currents, $I_{c,4\pi}/I_{c,2\pi}$ and a distorted SQI pattern.

*Model*— We model the Josephson junction depicted in Fig. 1 by means of the proximitized BHZ Hamiltonian [21] in the absence of Rashba or Dresselhaus spin-orbit coupling. Even though spin-orbit couplings are naturally present in QSH insulators, such as HgTe [22, 23], here, we assume they will not modify fundamentally the backscattering mechanism between ABS. This approximation simplifies our numerical calculations since we can express the resulting Bogoliubov de Gennes (BdG) Hamiltonian in a reduced basis, that is, $H = 1/2 \int \mathrm{d}r^2 \Psi^\dagger(\mathbf{r}) \mathcal{H} \Psi(\mathbf{r})$, with

$$\mathcal{H} = \begin{pmatrix} \mathcal{H}_e & \Delta \mathbb{1}_{2\times 2} \\ \Delta^* \mathbb{1}_{2\times 2} & -\mathcal{H}_e^* \end{pmatrix}, \qquad (2)$$

and the field operator

$$\Psi(\mathbf{r}) = \left[ c_{E\uparrow}(\mathbf{r}), c_{H\uparrow}(\mathbf{r}), c_{E\downarrow}^\dagger(\mathbf{r}), c_{H\downarrow}^\dagger(\mathbf{r}) \right]^T, \qquad (3)$$

written in the reduced subspace for BdG electrons (holes) with spin $\uparrow$ ($\downarrow$). Here, $c_{a\sigma}^{(\dagger)}(\mathbf{r})$, destroys (creates) an electron with orbital $a = E, H$, spin $\sigma = \uparrow, \downarrow$ at position $\mathbf{r}$.

The electronic part of the BdG Hamiltonian is given by the BHZ model $\mathcal{H}_e = \varepsilon(\hat{k}) + M(\hat{k})\sigma_z + A\left(\hat{k}_x \sigma_x - \hat{k}_y \sigma_y\right)$, with $\varepsilon(\hat{k}) = C - D\hat{\mathbf{k}}^2$, $M(\hat{k}) = M - B\hat{\mathbf{k}}^2$ and $\hat{\mathbf{k}} = -i\hbar\nabla_{\mathbf{r}}$,

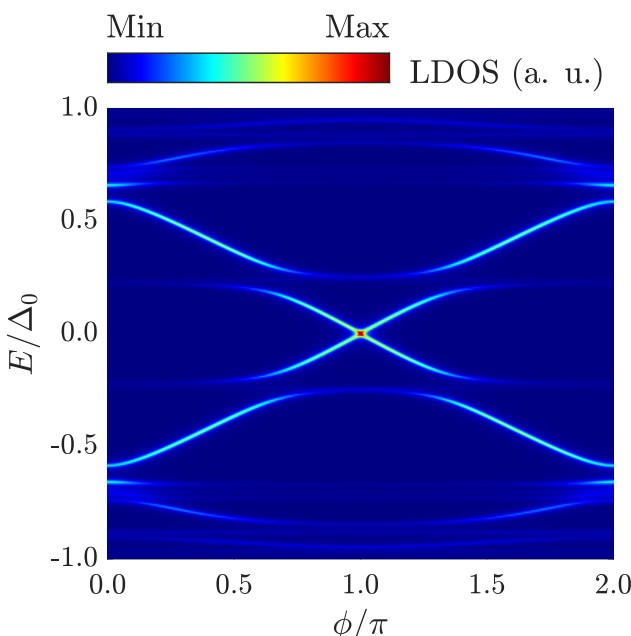

FIG. 2. Andreev bound states for a N'SNSN' Josephson junction with $L_s = 0.38\,\mu$m and $\xi_s = 0.19\,\mu$m, $L_{N'} = 0.5\,\mu$m. Avoided crossings coincide with the position of the energy levels at $E \approx 0.25\Delta_0$ and $0.65\Delta_0$.

with the Pauli matrices $\sigma$ operating on the orbital degree of freedom $(E, H)$ and parameters detailed in Ref. [24]. In the BHZ model, the mass $M$ opens a gap in the semiconductor spectrum, yielding a trivial insulating gap for $M > 0$ and a non-trivial one for $M < 0$, with the emergence of helical edge states. Moreover, we define the superconducting leads by setting finite values to $\Delta(\mathbf{r})$, with a constant value in the $y$-direction and a step function in the $x$-direction, with $\Delta(\mathbf{r}) = \Delta_0 \exp(is\phi/2)$ within the ranges $L_N/2 + L_s \geq |x| \geq L_N/2$ and zero otherwise.

Using standard finite difference methods, we discretize the Hamiltonian (2) replacing the coordinate $\mathbf{r} \to (i, j)a$, with $i, j \in \mathbb{Z}$ and the lattice constant $a = 5$ nm. Then, we model the dimensions and length scales of the Josephson junction guided by the physical phenomena taking place in typical experimental setups. Namely, a width W so large such that the overlap between opposite edges [25] and cross-Andreev processes are negligible [26]. To this aim, we use W $= 1\,\mu$m, $M = -10$ meV and $\Delta_0 = 0.6$ meV, yielding a coherence length $\xi_s \approx 190$ nm and a QSH edge localization length $l_{\mathrm{loc}} \approx 23$ nm [25]. Accordingly, we use the superconducting leads thickness of $L_s = 380$ nm, such that the QSH edges placed on N and N' are coupled, see Fig. 1.

*Coupled ABS*— Once we have set the length scales of the system, we move on and study the Andreev bound state spectrum by computing the local density of states (LDOS) along a section on the normal central part of the Josephson junction as a function of $\phi$. Due to the large size of the discrete system [$\dim(\mathcal{H}) \sim 10^5$], we make use

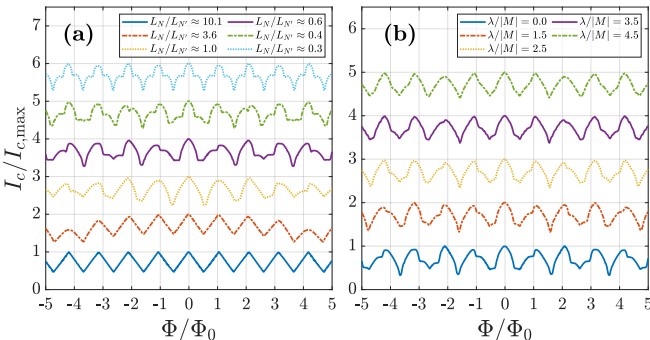

FIG. 3. The SQI pattern for Josephson junctions with the same parameters as in Fig. 2 but for (a) different $L_{N'}/a = 7, 20, 70, 120, 170, 220$ and (b) in the presence of disorder with increasing disorder strengths $\lambda$ for $L_{N'}/a = 100$. For better visibility, the SQI curves are shifted upwards by a constant $\Delta I_c = I_{c,max}$.

of recursive Green's functions (GFs) methods [27] and express the LDOS in terms of the imaginary part of the advanced GF along the normal part N, namely

$$\text{LDOS}(E) = \frac{1}{\pi}\text{Im}\sum_{x=1}^{n}\text{Tr}_W\{G^a(x,x,E)\}, \qquad (4)$$

with the GFs evaluated at position $x$ and energy $E$, along the normal part section with length $na$. This GF is represented by a $4\bar{W} \times 4\bar{W}$ matrix, with $\bar{W} \equiv W/a = 200$, see further details in Ref. [28].

In Fig. 2, we show a representative example of the energy-phase relation for the extended Josephson junction N'SNSN' with $L_s = 0.38\,\mu$m. Here, we can observe the emergence of avoided level crossings around the phase-independent ABS positions, i.e. $E_0 \approx 0.25\Delta_0$ and $E_1 \approx 0.65\Delta_0$. The size of these gaps scales with the geometric factor $\sim \Delta_0 \exp(-L_s/\xi_s) \approx 0.1\Delta_0$, which determines the transparency of the superconducting lead [29]. Naturally, in the limit of $L_s \gg \xi_s$, we recover the ABS spectrum touching the quasicontinuum.

The presence of avoided level crossings isolates energetically the lowest $4\pi$-periodic ABS from the rest of the $2\pi$-periodic spectrum, suppressing the emission of particles into the quasicontinuum if the driving is adiabatic enough [12–14, 16]. In this coupled scenario, the topological protection survives and gives rise to a finite $4\pi$-periodic supercurrent with $I_{c,4\pi} \approx eE_0/2\hbar$ for the total even parity $(p_{\text{top}}, p_{\text{bottom}}) = (0,0)$ and $(1,1)$ and cancels (in the absence of a magnetic flux) for the odd parity $(p_{\text{top}}, p_{\text{bottom}}) = (0,1)$ and $(1,0)$ [30–32].

There are remarkable differences between the backscattering process studied here and those resulting from either the direct overlap between the QSH wavefunctions [33] or a mass-inverted setup [31, 34]. There, the ABS spectrum develops gap openings at every time-reversal symmetry point $\phi = 0$ and $\phi = \pi$. Only for

some specific "sweet spots" in the parameter regime, one can find crossings at $\phi = \pi$ and avoided level crossings at $\phi = 2n\pi$, with $n \in \mathbb{Z}$. However, these "sweet spots" turn into anticrossings immediately, by changing any parameter of the system like the gate voltage, showing a lack of robustness for any practical purposes. In contrast, in the extended N'SNSN' junction, the gap openings are set by the geometry of the junction, yielding an unperturbed zero energy crossing as long as the discrete level $E_0$ is away from zero energy, i.e. $E_0 \gg \Delta_0 \exp(-L_s/\xi_s)$.

*Signatures of the backscattering process* — We shift our focus to studying signatures of the coupling between phase-dependent and -independent ABS. Since the backscattering mechanism introduced here modifies drastically the spatial distribution of the supercurrent, we expect already a modified critical current $I_c$ as a function of an external magnetic flux $\Phi = BWL_N$, also known as the SQI pattern [35].

The change in the SQI pattern is present for parity constrained or unconstrained scenarios. Thus, we calculate $I_c$ from the maximum of the equilibrium supercurrent $I_c = \text{Max}_\phi\{I(\phi)\}$, with

$$I(\phi) = \frac{e}{\hbar^2}\int dE\,\text{Tr}_W\left\{[V_{LR}G_{RL}^{+-}(E) - V_{RL}G_{LR}^{+-}(E)]_e\right\}, \tag{5}$$

where $G_{RL}^{+-}(E) \equiv G^{+-}(x_0 - a, x_0 + a, E)$ are the equilibrium lesser GFs evaluated at $L = x_0 - a_x$ and $R = x_0 + a_x$ positions, with $x_0$ placed on the central normal part of the junction. Besides, $V_{LR/RL}$ couples different layers of the discretized Hamiltonian. Here, we take the trace over the electron part of the GFs and the width W of the junction.

In Fig. 3(a), we study the impact of considering different ratios $L_{N'}/L_N$ on the SQI pattern. Thus, we increase $L_{N'}$ in steps of $50\,a$ and show the normalized $I_c/I_{c,\text{max}}$ for an increasing $L_{N'}$ from bottom to top. Here, $I_{c,\text{max}}$ is the maximum value of $I_c$ over $\Phi$ for a given set of parameters. For clarity, we shift vertically each curve as we increase $L_{N'}$. We can observe that deviations of the sinusoidal character of the SQI pattern arise for $\Phi/\Phi_0 \sim L_N/L_{N'}$, e.g. for $L_N/L_{N'} \approx 3.6$, we observe changes for $\Phi/\Phi_0 \geq 3$, as the area enclosed by the N' regions needs a comparable magnetic flux. For larger $L_{N'}$, the SQI pattern changes its periodicity for smaller magnetic flux, developing erratic patterns or periodic ones, depending on the relative fluxes threading the N and N' regions. Note that introducing different lengths for the left and right $L_{N'}$ can lead to an even more erratic pattern, since in this situation, there is an extra periodicity coming into play [28].

The presence of potential disorder on the N' regions broadens the energy spectrum of the phase-independent ABS [28]. Consequently, the coupling between phase-dependent and -independent ABS becomes effectively suppressed. To see its influence on the SQI pattern, we

introduce a random local potential on the sites placed at the N' regions within the values $[-\lambda, \lambda]$. In Fig. 3(b), we can observe that the features distorting the SQI pattern are smoothed out for increasing $\lambda$, resulting from the broadening of the phase-independent ABS. Indeed, for $\lambda \lesssim 3|M|$, we observe numerically that the ABS become broader, but nevertheless, well-formed independently of $p_{N'}$ [28]. For larger disorder strengths $\lambda > 4|M|$, the topological protection of the QSH edges is effectively removed since QSH edge states can now hybridize with the bulk states. Consequently, the phase-independent ABS do not form, yielding approximately the conventional picture of QSH Josephson junction [36].

To have a direct proof of the phase-independent ABS participation, we propose to selectively remove the topological character by means of a finite magnetic flux. Due to the different areas in the N and N' regions, a finite magnetic flux shifts relatively the position of phase-independent and phase-dependent ABS. Indeed, the former become

$$E_n = \frac{\hbar v_F}{p_{N'}} \pi (n + 1/2 + \alpha \Phi/\Phi_0), \qquad (6)$$

with $n \in \mathbb{Z}$ and $\alpha = L_{N'}/L_N$. Hence, setting $E_n \approx 0$ at the maximum of the SQI, namely,

$$\Phi/\Phi_0 \approx -\text{int}\{(n + 1/2)/\alpha\}, \qquad (7)$$

we can gap out the MBS, removing the fractional Josephson effect. This change can be probed by means of the Shapiro experiment or the Josephson radiation.

To see this phenomenon explicitly, we represent in Fig. 4(a) the SQI pattern together with the integrand of Eq. (5), henceforth called $I(\phi_c, E)$, as a function of the energy $E$ and $\Phi/\Phi_0$ in panel (b) [37]. Here, we can observe two types of (roughly) periodic contributions, one with an ∫-shape and period $\Phi/\Phi_0 \approx 1$, which results from the ABS localized at the SNS region, and a linear dispersion, with period $\alpha\Phi/\Phi_0 \approx 1$, which arises from the phase-independent ABS. Remarkably, the largest distortion on the SQI pattern (a) coincides with the presence of phase-independent ABS at zero energy, see vertical red dashed lines. Moreover, when both contributions coincide in energy and flux, they develop a minimum resulting from the hybridization process between them. As we have anticipated, when both contributions coincide around zero energy around an integer value of $\Phi/\Phi_0$, the topological state turns into a trivial one. We show the resulting ABS as a function of $\phi$ for $\Phi/\Phi_0 = 1$ in Fig. 4(b). We observe a gap opening around zero energy, yielding a $2\pi$-periodic ABS. Note that the lack of particle-hole symmetry in the spectrum originates from the combined reduction of the Nambu basis and the presence of a finite magnetic flux. We recover the particle-hole symmetry when repeating the calculation with the complete basis, see [28].

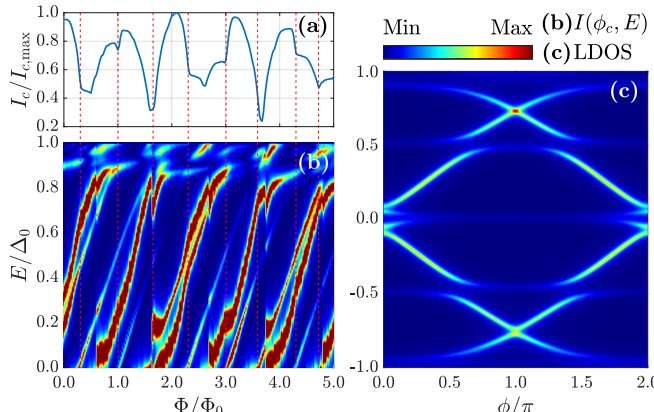

FIG. 4. Panel (a): critical current vs magnetic flux. Panel (b): integrand of Eq. (5) as a function of energy and $\Phi/\Phi_0$, for $\phi = \phi_c$ fulfilling $I_c = I(\phi_c)$, with the same parameters as in Fig. 2. Panel (c): Andreev bound spectrum as a function of $\phi$, with $\Phi$ fulfilling the condition (7), where the phase independent ABS hybridizes at zero energy with the phase-dependent ABS. We set $L_s = 0.33\,\mu$m to make the gaps more visible.

*Discussion* — Using a non-interacting tight-binding model to describe an extended quantum spin-Hall Josephson junction N'SNSN' under time-reversal symmetry, we design a backscattering mechanism that isolates energetically a $4\pi$-periodic contribution from the rest of the spectrum. In this geometry, additionally to the phase-dependent Andreev bound states localized at the central SNS junction, extra phase-independent Andreev bound states form at the edges of the N'S regions with discrete energies $E_n = \pi\hbar v_F(n + 1/2)/p_{N'}$ determined by the perimeter of the N' part $p_{N'}$. Thus, in a scenario with transparent enough superconducting leads, these additional ABS mediate a backscattering mechanism between opposite edges on the SNS junction, resulting in the appearance of avoided level crossings when both types of Andreev bound states are at resonance, yielding a measurable $4\pi$-periodic gap $\Delta_{4\pi,\text{eff}} = E_0 = \pi\hbar v_F/2p_{N'} \sim 0.57 - 0.11\,\text{meV}$, for $p_{N'} \sim 1 - 5\,\mu$m. The resulting ABS remains topological [30, 31] and can be probed, for example, by means of the Shapiro experiment or the Josephson radiation if the driving is adiabatic enough [12–14, 16]. Furthermore, we have tested the stability of the phase-independent ABS against the presence of potential disorder finding a robust behavior for disorder strengths larger than the topological gap, i.e. $\lambda \lesssim 3|M|$ and can extend on the order of several microns.

We predict signatures of this backscattering mechanism to be present in the SQI pattern, which becomes distorted due to the additional trajectories enclosing the N'S regions. These trajectories gather an additional magnetic flux and hence, introduce a new periodicity into the sinusoidal $\Phi_0$-periodic SQI pattern, which develops local maxima and minima on the magnetic flux scale of

$\sim (L_N/L_{N'})\Phi_0$. Moreover, due to the difference between $L_N$ and $L_{N'}$, we can use the magnetic flux to tune phase-independent Andreev bound states towards the phase-dependent ones. We find that when these two types of ABS become close to zero energy, we remove the $4\pi$-periodicity selectively, and therefore, it can be used as a control knob to switch on and off the fractional Josephson effect.

We believe that the use of this backscattering mechanism can contribute decisively to the design of topological Josephson junctions with $4\pi$-periodic ABS energetically isolated from the quasicontinuum under time-reversal symmetry. Moreover, the physics described here is not restricted to the given N'SNSN geometry. Indeed, analogous ideas can be extended onto a single edge of two topological Josephson junctions in series, i.e. the SNSN'SNS junction. Here, the central N' region develops phase-independent ABS when the two inner (outer) superconductors share the same phase difference.

*Acknowledgments* — We acknowledge stimulating discussions with B. Trauzettel, E. Bocquillon, E. M. Hankiewicz, N. Traverso Ziani, M. Stehno and L. W. Molenkamp. F.D. and P.R. gratefully acknowledge funding by the Deutsche Forschungsgemeinschaft (DFG, German Research Foundation) under Germany's Excellence Strategy – EXC-2123 QuantumFrontiers – 390837967.

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

# Supplemental Material for "Interedge backscattering in time-reversal symmetric quantum spin Hall Josephson junctions"

## S1. DETAILS ON THE TRANSPORT FORMALISM

We are interested in modelling a discrete $2D$ material, therefore, we apply the tight-binding discretization method on the Hamiltonian in Eq. (2) by replacing the continuous momentum operators $\hat{k}_{x,y} = -i\partial_{x,y}$ by their discretized versions in the Hamiltonian of Eq. (2)[38]:

$$\partial_{x/y}\Psi(x,y) \approx \frac{1}{2a_{x/y}}\left(\Psi_{j_{x/y}+a_{x/y}} - \Psi_{j_{x/y}-a_{x/y}}\right) \tag{1}$$

$$\partial^2_{x/y}\Psi(x,y) \approx \frac{1}{a^2_{x/y}}\left(\Psi_{j_{x/y}+a_{x/y}} - 2\Psi_{j_{x/y}} + \Psi_{j_{x/y}-a_{x/y}}\right), \tag{2}$$

where $a_{x,y}$ are the lattice constants of the two-dimensional lattice and $j_{x,y}$ is the site index in $x$ and $y$ direction. Until specified otherwise, we will use $a = a_x = a_y = 5\,\text{nm}$ as the lattice constant. The corresponding tight-binding Hamiltonian thus takes the following form

$$\mathcal{H} = \sum_{n=1}^{L}\sum_{m=1}^{W} \mathcal{H}^0_{\substack{nn\\mm}} + \mathcal{V}_{\substack{n,n+1\\m,m}} + \mathcal{V}_{\substack{n,n\\m,m+1}} + \mathcal{V}^\dagger_{\substack{n,n+1\\m,m}} + \mathcal{V}^\dagger_{\substack{n,n\\m,m+1}}, \tag{3}$$

where L and W represent the length and width of the junction and the indices $n$ and $m$ represent the spatial dimensions $x$ and $y$, respectively. $\mathcal{H}^0$ is the on-site Hamiltonian and $\mathcal{V}$ describes the hoping between neighbouring sites.

To efficiently treat the tight-binding Hamiltonian of Eq. (3), we employ a standard recursive Green's method only for the $x$-direction of the Hamiltonian, i.e. we do not perform the sum over $L$ but instead grow the lattice recursively with Green's functions. To see what this means, we first rewrite Eq. (3) as

$$\mathcal{H} = \sum_{n=1}^{L} H^0_{nn} + V_{n,n+1} + V^\dagger_{n,n+1}, \tag{4}$$

$$H^0_{nn} = \sum_{m=1}^{W} \mathcal{H}^0_{\substack{nn\\mm}} + \mathcal{V}_{\substack{n,n\\m,m+1}} + \mathcal{V}^\dagger_{\substack{n,n\\m,m+1}}, \tag{5}$$

$$V_{n,n+1} = \sum_{m=1}^{W} \mathcal{V}_{\substack{n,n+1\\m,m}}, \tag{6}$$

where $H^0_{nn}$ describes the on-site Hamiltonian of the $n$-th stripe of $\dim(H^0_{nn}) = 4\bar{W} \times 4\bar{W}$ and $V_{n,n+1}$ describes the hopping from the $n$-th stripe to the $n+1$-th stripe. Since the system has a homogeneous hopping in the $x$-direction, we simplify the notation $V_{LR} \equiv V_{n,n+1}$ and $V_{RL} \equiv V^\dagger_{n,n+1}$. Using this notation, we find the perturbed retarded/advanced Green's function $G^{r/a}_n$ by the recursive scheme [27]

$$G_{nn} = \left[g^{-1}_{nn} - V_{RL}G^L_{n-1,n-1}V_{LR} - V_{LR}G^R_{n+1,n+1}V_{RL}\right]^{-1}, \tag{7}$$

$$G^L_{nn} = \left[g^{-1}_{nn} - V_{RL}G^L_{n-1,n-1}V_{LR}\right]^{-1}, \tag{8}$$

$$G^R_{nn} = \left[g^{-1}_{nn} - V_{LR}G^R_{n+1,n+1}V_{RL}\right]^{-1}, \tag{9}$$

where we lightened the notation by omitting the superscript $r/a$. $g^{r/a}_{nn} = [(E \pm i\eta)\mathbb{1}_{4\bar{W}\times 4\bar{W}} - H^0_{nn}]^{-1}$ is the unperturbed Green's function for the $n$-th stripe. Furthermore, we set $G^R_{00} = G^L_{00} = \mathbb{0}$, with $\mathbb{0}$ as the zero-matrix, since in the first iteration we only have a single site, which is by definition unperturbed $G^R_{11} = G^L_{11} = g_{11}$. We obtain the LDOS by taking the imaginary part of the perturbed advanced Green's function

$$\text{LDOS}(E) = \frac{1}{\pi}\text{Im}\sum_{n=1}^{N}\text{Tr}_W\{G^a_{nn}\}, \tag{10}$$

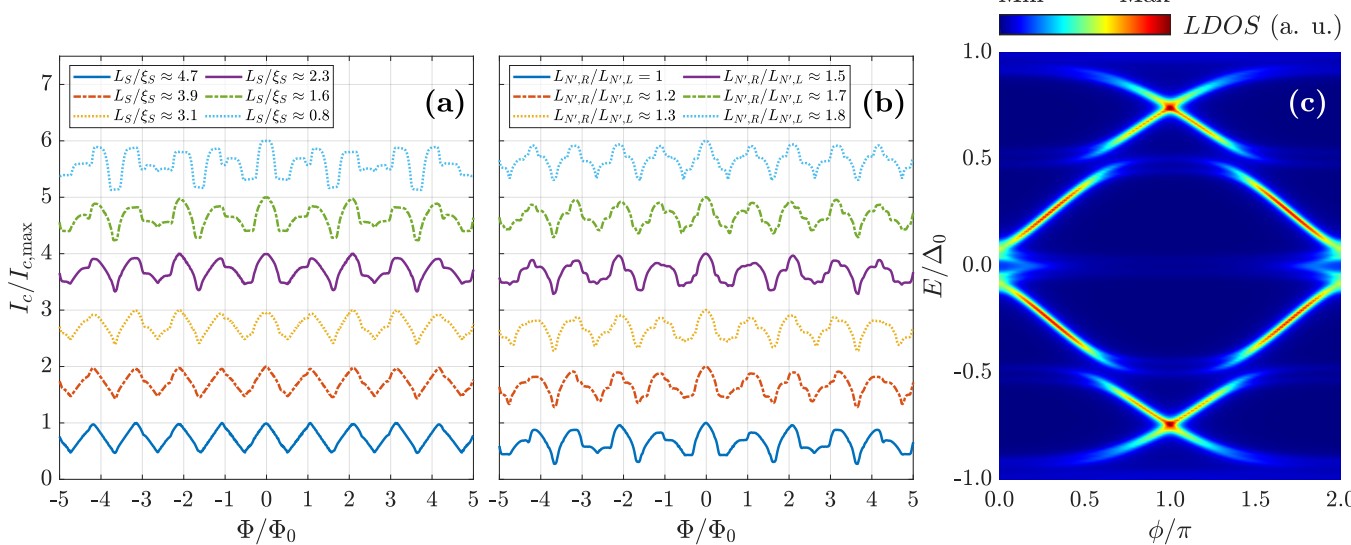

FIG. S1. Panel (a): critical current versus magnetic flux for (a) different lengths of the superconductor $L_S/a = 180, 150, 120, 90, 60, 30$ and (b) different lengths of the external parts. We fix $L_{N',L}/a = 120$ and increase the length of the right part to $L_{N',R}/a = 140, 160, 180, 200, 220$. For visibility, the SQI curves are shifted upwards a constant $\Delta I_c = I_{c,max}$. Panel (c): The Andreev bound state spectrum for the full BHZ Hamiltonian with both spin degrees of freedom.

where $\text{Tr}_W$ is the trace with respect to the width W and $N$ is the number of sites we calculate the LDOS on and which can be chosen to be smaller than $L$.

To obtain the supercurrent of the Josephson junction, we imagine splitting the junction along the $x$-direction in a left $(L)$ and a right $(R)$ part. The current then takes the following form

$$I(\phi) = \frac{e}{\hbar^2} \int_{-\infty}^{+\infty} dE \ \text{Tr}_W \left\{ [V_{LR} G_{RL}^{+-}(E) - V_{RL} G_{LR}^{+-}(E)]_e \right\}, \tag{11}$$

where, in equilibrium, the lesser Green's functions $G^{+-}$ simplify to

$$G_{LR}^{+-}(E) = [G_{LR}^a(E) - G_{LR}^r(E)] f(E), \tag{12}$$
$$G_{RL}^{+-}(E) = [G_{RL}^a(E) - G_{RL}^r(E)] f(E), \tag{13}$$

with $f(E)$ as the equilibrium Fermi-Dirac distribution. To make use of Eq. (11) we relate the Green's functions of Eq. (12) and (13) to the surface Green's functions of Eq. (8) and (9) as follows

$$G_{LR}^{r/a} = \mathcal{G}_{LL}^{r/a} V_{LR} G_{RR}^{r/a}, \tag{14}$$
$$G_{RL}^{r/a} = G_{RR}^{r/a} V_{RL} \mathcal{G}_{LL}^{r/a} \quad \text{and} \tag{15}$$
$$G_{RR}^{r/a} = [(\mathcal{G}_{RR}^{r/a})^{-1} - V_{RL} \mathcal{G}_{LL}^{r/a} V_{LR}]^{-1}, \tag{16}$$

where $\mathcal{G}_{LL/RR}^{r/a}$ represents the retarded/advanced surface Green's function of the left/right lead, i.e. $\mathcal{G}_{LL/RR} = G_{nn}^{L/R}$. Furthermore, we made use of the discretized form of the Dyson equation

$$\langle i|G|j \rangle = \langle i|G + \mathcal{G}VG|j \rangle \tag{17}$$
$$\Leftrightarrow \quad \mathcal{G}_{ij} = G_{ii}\delta_{ij} + G_{ii}V_{i,i-1}G_{i-1,i} + \mathcal{G}_{ii}V_{i,i+1}G_{i+1,j}. \tag{18}$$

## S2.  ADDITION TO THE SQI PATTERN

In addition to Fig. 3, we investigate the SQI pattern resulting from varying $L_s$, which controls the coupling strength between the phase-dependent and independent ABS, depicted in Fig. S1 (a). We set $L_{N'} = 0.6 \, \mu m$ and reduce $L_s$ in

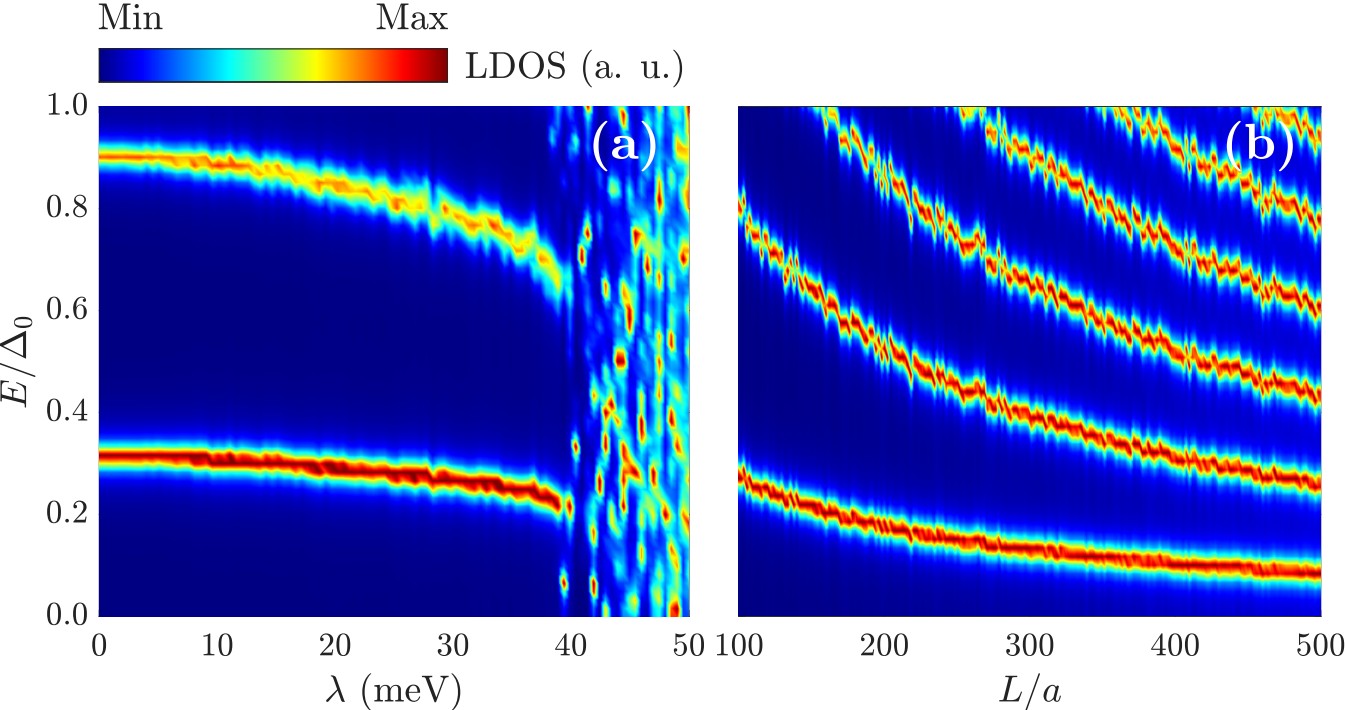

FIG. S2. Panel (a): The energy of the phase-independent ABS as a function of the disorder strength $\lambda$. We set $L_S = \infty$ and the length and width of the normal part to $L = W = 0.5\ \mu$m, respectively. In this plot, we averaged over 20 disorder configurations. Panel (b): The energy of the phase-independent ABS as a function of the length of the normal part $L$, with $a$ as the lattice constant introduced in the main part of the text. We set $\lambda = 35$ meV.

steps of $0.15\ \mu$m from $L_s = 0.9\ \mu$m to $L_s = 0.1\ \mu$m. For clarity, we have shifted the curves upwards for a decreasing $L_s$. For $\xi_s/L_s \ll 1$, we observe that the SQI pattern exhibits the typical sinusoidal pattern with periodicity $\Phi_0$, denoting the lack of coupling to the N' parts. As we decrease $L_s$, the superconducting leads become more transparent, enhancing the effective coupling between opposite edges. The resulting SQI pattern develops local maxima and minima on top of the sinusoidal pattern, and turns into an erratic pattern for $\xi_s/L_s > 1$. Moreover, we note that the non-normalized $I_c$ becomes more suppressed by reducing $L_s$ (not shown), since the avoided level crossings become more dominant, resulting in a flatter ABS spectrum.

In Fig. S1(b) we consider the scenario of a different left and right $L_{N'}$ parts, which introduces additional periodicities relative to panel 3(b). Due to the period mismatch of the left and right external parts, the SQI exhibits even more erratic patterns, compared to Fig. and Fig. S1 (a). In general, it is therefore difficult to predict a given periodicity. This is in contrast to the SQI patterns resulting from the direct (energy-independent) coupling between opposite edges, which can lead to an even-odd effect [34, 36, 39–42].

In the main text, we consider a single spin-degree of freedom of the original BHZ Hamiltonian. As a consequence, the system is not particle-hole symmetric in the presence of a magnetic flux, as shown in Fig. 4. However, we can restore the particle-hole symmetry upon including the second spin degree of freedom, as shown in Fig. S1 (c).

## S3. DISORDER

To investigate the effects of disorder on the transport properties of the Josephson junction, we add $\mathcal{H}_{Dis} = \lambda s \sigma_0 \otimes \tau_z$ to the Hamiltonian $\mathcal{H}$ of Eq. (2), where $\lambda$ is the disorder strength, $\tau$ represents the particle-hole degree of freedom and $s$ takes on random values between $-1$ and $1$ for each lattice site. Here, we restrict the analysis to a single external part attached to a semi-infinite superconductor [43]. In Fig. S2 (a) we observe that the phase-independent ABS are stable in the presence of disorder up to a critical strength of $\lambda_c \approx 4|M|$. Furthermore, we note that as $\lambda$ increases, the energy of the ABS diminishes, since disorder forces the particles to make a detour. Disorder strengths larger than the bulk gap can couple the top and bottom edges at every spatial point of the junction, lifting the time-reversal protection of the states.

In Fig. S2 we show that for a high disorder strength $\lambda = 35\,\text{meV}$ the ABS remain stable upon increasing the length of the normal part. Adding more sites to the QSHI allows for more scattering events of the helical edge with disordered sites, however, as long as the disorder strength stays below the critical value of $\lambda_c$ the ABS obey Eq. (6) up to minor distortions, resulting from a slightly altered perimeter. As previously mentioned, we also observe that the disorder has a progressively greater impact on the higher ABS.