# Peer review of "Interedge backscattering in time-reversal symmetric quantum spin Hall Josephson junctions"

_SciPost_

## Round 1 · Referee Report · Antonio Manesco (Referee 1) · 2024-12-14

Strengths

1 - The research question and the proposed solution are clear.
2 - The manuscript is self-contained and the authors provide an intuitive semiclassical picture whenever possible.
3 - The description of the numerical modeling is sufficiently detailed.

Weaknesses

1 - It is unclear if the proposal works in a more realistic setting, e.g. disorder and spatial-dependent Hamiltonian (see report).
2 - Code and data are not available. Although not required for acceptance, I think it is always helpful to ensure reproducibility.

Report

The authors propose a device to prevent the coupling of Andreev bound states with quasicontinuum to observe $4\pi$-periodic Josephson effect. The normal region in their devices consists of a quantum spin Hall insulator (QSHI), focusing on HgTe. The material is proximitized by two superconducting leads, forming a N'SNSN' structure. Because the inner normal region is sandwiched by two superconductors, the Andreev spectrum in this middle island is phase-dependent. The outer islands have a phase-independent spectrum. If the lengths of the superconducting regions are $\lesssim \xi$, the Andreev bound states in different islands couple, opening a gap in the spectrum. This gap allows driving the phase adiabatically such as the lowest-energy Andreev bound state never merges with the continuum. The authors compute the SQI pattern showing a $4\pi$-periodic Josephson effect insensitive to disorder below the gap size.

I am keen to recommend the publication of this work. However, I would like the authors to clarify a few points before reaching my final decision.

  1. Is there a fundamental difference between coupling with the quasicontinuum and coupling with the Andreev bound states at the outer islands? I would naively imagine that the outer dots are a superconductor with a discrete density of states. Why is that different from a quasicontinuum? Is it because these isolated Andreev levels have fixed fermionic parity?
  2. The authors discuss the protection of $4\pi$-periodicity using a notation for fermionic parity $(p_{\text{top}}, p_{\text{bottom}})$, but I failed to follow what the "top" and "bottom" refer to. I think it is helpful to explain this notation in the text.
  3. The device conception requires phase-independent Andreev bound states coupled to a phase-dependent one. Are the two outer regions needed? Would the same device work with a single outer normal region?
  4. The authors assume a constant Fermi velocity for their estimations. Is it reasonable even if accounting for band-bending effects from the proximity with the superconductor?
  5. The authors consider the effects of disorder in the normal region. What about disorder in the proximitized superconducting regions? I would expect this type of disorder to broaden the spectral lines close to the anticrossings. Would that affect the protection of the $ 4\pi$ periodicity?
  6. What do you expect to happen if there is a Fermi level mismatch between the normal and superconducting regions? Would that change the coupling between the Andreev states?

Requested changes

1 - Account for effects of disorder and Fermi energy mismatch.
2 - Clarify the difference between isolated Andreev bound states and quasicontinuum in phenomenology.
3 - Introduce the notation for fermionic parity.

Recommendation

Ask for minor revision

---

## Round 1 · Referee Report · Anonymous (Referee 2) · 2024-12-19

Strengths

  • Good motivation (quasicontinuum decoupling in topological Josephson junction)
  • Intriguing geometry combining two interfering current paths, with characteristics tuneable through flux

Weaknesses

  • Simplistic modelling for the superconductor and the continuum (see report)
  • Easier alternatives exist (e.g. quantum dots, see report)
  • Value of the SQI pattern simulations somewhat unclear within the scope of the motivation
  • Why not actually show Shapiro-step simulations instead of SQI?

Report

The work presents a clever idea for manipulating a Josephson junction spectrum of based on interference of two kinds of paths in a QSH junction. The essence is the coupling of phase-independent states with phase-dependent Andreev states. This coupling occurs through non-local processes across the superconducting contact, so that their length is critical, and non-tunable. A much easier alternative exists, I believe: just couple a quantum dot to the edges, and tune its spectrum electrically. Would this work? Any advantage of the outer QSH state idea?

The main motivation is to decouple the Andreev states from the continuum. However, the continuum is very poorly modelled. Instead of an omega-dependent superconductor selfenergy (which would include a quasiparticle continuum above a parent gap), a simple omega-independent pairing is used. Hence the "quasicontinuum" is just the sparse state spectrum under the contact. Since the gap edge repulsion using a proper self-energy critically affects the amount of coupling of ABS to the continuum, this is a serious flaw of the analysis. Also, the method used for the simulations is based on Green's functions and energy integrals, so adding the correct self energy (ballistic or diffusive) should come at no extra computational cost.

Finally there is a considerable digression about SQI distortions that does not quite offer enough value to the paper in my opinion, and seems a bit of a tangent. Instead of SQI, the obvious thing to analyse would have been precisely Shapiro steps in the presence of the decoupling mechanism. The SQI results are at some points also a bit puzzling. For example, why does the large disorder SQI pattern resemble a SQUID-like pattern instead of a Fraunhofer-like pattern? According to the discussion, strong disorder should restore the "conventional SQI".

As a whole I think this paper deserves publication, but is perhaps better suited for Scipost Core, given the above limitations. I do not believe it offers the kind of "groundbreaking new results" that are required for Scipost Physics.

Requested changes

  • Consider LN´ = 0 and add a quantum dot coupled to one or both edges. Does it provide the same effect as a finite LN´? If not explain why. If it does, state it clearly as an alternative approach.

  • Redo simulations with an omega-dependent superconductor self-energy, with a proper quasiparticle continuum and gap edge

  • Clarify why large-lambda SQI looks like a SQUID (edge-dominated transport)

  • If at all possible, consider simulating Shapiro steps or Josephson radiation (optional)

Recommendation

Accept in alternative Journal (see Report)

---

## Editorial Decision

awaiting_resubmission